# A Comparative Analysis of Deep Learning Models for Automated Cross-Preparation Diagnosis of Multi-Cell Liquid Pap Smear Images

**DOI:** 10.3390/diagnostics12081838

**Published:** 2022-07-29

**Authors:** Yasmin Karasu Benyes, E. Celeste Welch, Abhinav Singhal, Joyce Ou, Anubhav Tripathi

**Affiliations:** 1Center for Biomedical Engineering, School of Engineering, Brown University, Providence, RI 02912, USA; yasmin_karasu@alumni.brown.edu (Y.K.B.); cel_welch@brown.edu (E.C.W.); 2Department of Computer Science and Engineering, I.I.T. Delhi, Hauz Khas, New Delhi 110016, India; abhinavsinghal@outlook.com; 3Department of Pathology and Laboratory Medicine, Alpert Medical School, Brown University, Providence, RI 02912, USA; joyce_ou@brown.edu

**Keywords:** cervical cancer, medical diagnostic imaging, neural networks, image classification, artificial intelligence

## Abstract

Routine Pap smears can facilitate early detection of cervical cancer and improve patient outcomes. The objective of this work is to develop an automated, clinically viable deep neural network for the multi-class Bethesda System diagnosis of multi-cell images in Liquid Pap smear samples. 8 deep learning models were trained on a publicly available multi-class SurePath preparation dataset. This included the 5 best-performing transfer learning models, an ensemble, a novel convolutional neural network (CNN), and a CNN + autoencoder (AE). Additionally, each model was tested on a novel ThinPrep Pap dataset to determine model generalizability across different liquid Pap preparation methods with and without Deep CORAL domain adaptation. All models achieved accuracies >90% when classifying SurePath images. The AE CNN model, 99.80% smaller than the average transfer model, maintained an accuracy of 96.54%. During consecutive training attempts, individual transfer models had high variability in performance, whereas the CNN, AE CNN, and ensemble did not. ThinPrep Pap classification accuracies were notably lower but increased with domain adaptation, with ResNet101 achieving the highest accuracy at 92.65%. This indicates a potential area for future improvement: development of a globally relevant model that can function across different slide preparation methods.

## 1. Introduction

Cervical cancer is a prominent gynecological cancer with 311,000 annual deaths globally [1]. In recent decades, the United States and various other countries have implemented standardized routine cervical cancer screening protocols utilizing the Papanicolaou test, or “Pap smear”, a staining method used to discern cancerous and precancerous cells via optical microscopy.

In traditional Pap sample collection workflows, a speculum is inserted, and the transformation zone of the cervix is scraped with a spatula. The cells are then applied to a glass slide and stained with a traditional protocol utilizing five separate stains in three separate solutions [2,3]. Microscopic examination is subsequently performed by cytotechnologists and, in cases where atypical cells require definitive diagnosis, board-certified pathologists.

The traditional Pap workflow has been advanced by recent technology developments, including the integration of minimally invasive sample collection approaches, such as cellular collection with a brush or broom-like device. The newly emerging standard of liquid preparations enables the collection of clearer images with less cellular obstruction when compared to traditional smeared slides.

Many different liquid pap instruments, staining protocols, and associated devices have emerged in the past several years. This creates significant heterogeneity in the visual appearance of cells in images, cellular densities, and distributions, even among preparations under the umbrella of “liquid Pap smears”. Accordingly, the image analysis component of the workflow has also been the target of numerous attempts at improvement.

Software such as CHAMP digital imaging software has been created for automated diagnosis of Pap smear images via image segmentation [4]. However, this type of software is expensive, limiting translation into resource-limited countries. 

A fair amount of research has been conducted in recent years on methods for applying machine learning algorithms to diagnose cervical cancer via cytological image analysis [4]. These methods can be used to perform automatic classification at the single-cell and multi-cell level, using binary (cancerous vs. not cancerous) and multivariate prediction (classified based on The Bethesda system) [5]. The Bethesda system classifies cells into multiple diagnostic categories, including (but not limited to) Negative for Intraepithelial Malignancy (NILM), Low Grade Squamous Intraepithelial Lesion (LSIL), High Grade Squamous Intraepithelial Lesion (HSIL), and Squamous Cell Carcinoma (SCC).

While machine learning has helped to streamline and expedite image analysis to a degree, certain limitations remain. Single-cell analysis and binary classification have major limits to clinical translational potential as they rely on expensive auto segmentation technologies and lack the specificity of the Bethesda system, respectively [6]. Additionally, manual feature extraction and image segmentation remain a major implementation challenge in many workflows [7,8]. However, a significant body of promising research has been conducted in recent years on optimized machine learning techniques for Pap smear and other cervical cancer diagnosis [4,9,10].

Deep learning could serve as a promising alternative, allowing for manual feature extraction and segmentation steps to be replaced with a larger and more complex feature set. Deep learning has been shown to outperform traditional machine learning on similar tasks [11]. Recent work has been conducted on using deep learning for functions such as cell image segmentation, classification, and ranking [11,12,13].

Recently, Hussain et al. created a liquid Pap smear image dataset using the SurePath method to prepare samples [14]. Hussain et al. subsequently demonstrated the utility of deep learning in the processing of these and other images [15]. However, while this represents the largest existing publicly available, Bethesda classified dataset of multi-cell liquid Pap smear images, it is nevertheless limited in size and unbalanced. Furthermore, the performance of a model trained against this dataset must still be assessed by classifying new, clinically obtained multi-cell images. 

Unbalanced datasets and models that are trained and tested with only one dataset are often thought to be limited in terms of their practical clinical utility. Herein, the clinical translational potential of deep learning will be further investigated by addressing these factors. Furthermore, the potential of deep learning models to classify different types of multi-cell liquid Pap preparations will also be investigated, to our knowledge, for the first time in the literature. We propose and test the specific technique of unsupervised domain adaptation as a solution to this cross-preparation classification problem, through application of Deep CORAL [16].

## 2. Materials and Methods

### 2.1. Novel Clinical Image Collection

Sixty-eight images were obtained from a previously deidentified and anonymized cytopathology teaching slide collection built by gynecologic pathologists at Women and Infants Hospital of Rhode Island and Brown University Alpert School of Medicine. These images were used to create the bulk of the clinical testing dataset. All images were manually acquired from diagnostically verified ThinPrep Pap test (Hologic, Inc., Marlborough, MA, USA) slides prepared using clinical standard operating procedures. 

All microscopic images were captured with a 40× objective, 10× eyepiece, 0.5× C-mount adapter, and an Excelis HD color microscopy camera. Images were taken using matched interrogation areas with identical stated image size and resolution to the Hussain et al., database (2048 × 1536 pixels, 400× magnification). Image segments were then reviewed again by two pathologists and classified as follows: 72 NILM, 56 LSIL, 48 HSIL, and 54 SCC. The resulting images were then tested using the classification algorithms (Figure 1).

### 2.2. Data Augmentation

Deep learning models rely on sufficiently large datasets to effectively learn the features of complex classes. Training a model on a large number of samples allows it to fine-tune its weights to more accurately predict each class and differentiate between classes. The Hussain et al. dataset only contains 963 images separated into 4 unbalanced classes. Data augmentation is a common approach used to increase the size and balance class distribution of similar datasets [17]. 

Additional images for training were created using the ImageDataGenerator function from the Keras library (Table 1). Images randomly selected from the dataset were duplicated and modified via randomized rotation and flipping. As the NILM class was the largest, the 3 other classes were augmented to an equal size. This technique prevents models from prioritizing the prediction of larger classes such as NILM and consistently underpredicting SCC due to its limited data size. 

Using a broader range of variations would have enabled the dataset to become larger, however, due to the shape and level of detail on the edges of images, excessive rotation decreased the quality of the images and added variations in the appearance of the cells that do not naturally occur, thereby reducing clinical viability.

Hence, the augmentation was limited to a size that avoids duplicates and maintains the integrity of images. The augmented dataset contains 2452 images with 613 images in each class. For all model training, the data has been split into 65:15:20% for training, validation, testing. This is to ensure that there is sufficient data to adequately train and test the models. The data was split using a randomized shuffle so there is an equal distribution of each class in each of the 3 groups.

Deep learning models are ideally trained on large datasets [17]. Despite augmentation, the dataset remained smaller than ideal for deep learning. A common technique used to minimize overfitting caused by smaller datasets is transfer learning [18]. This approach is also used to avoid the computational cost of training a model from scratch [19]. Transfer models are publicly available and trained on millions of classified images. 

To fine-tune a transfer model to classify pre-cancerous cells, the final fully-connected layer of the model was isolated, replaced by a layer with 4 output nodes, and re-trained by adjusting its weights. This was facilitated by continuing backpropagation using the Hussain et al. dataset.

24 transfer learning models were trained using Adam optimizer for 25 epochs with a batch size of 16 at a learning rate of 0.001 to ensure adequate training time while preventing overfitting. Although there were more models within the set of publicly available transfer learning models, not all of them were applicable to the dataset used in this paper and they were thus excluded from consideration.

Prior research has demonstrated that Alexnet [20], Vggnet (Vgg-16, Vgg-19) [21], Googlenet [22], and ResNet (ResNet50, ResNet101) [23] are the best performing models for liquid Pap classification [16]. However, by training all the relevant, publicly available models, a new set of best performing models was determined using accuracy and F1 score as the primary metrics.

These Image classification models with transfer learning use 4–60 million trainable parameters to classify images (Table 2). While transfer learning eliminates the need to train models from scratch, models with fewer trainable parameters enhance generalization and avoid overfitting [24]. Additionally, too many trainable parameters risk overfitting which causes them to be unable to generalize to novel images.

### 2.3. Ensemble

Ensemble methods depend on combining a group of independently trained models into a single model in order to make a prediction. Each model within the group makes a prediction that acts as a vote, and the collective votes of the group become the prediction of the ensemble method. 

There are a variety of voting methods that can be used. Two commonly used methods are taking the average of the votes and majority voting [25]. Since the labels of this dataset are discrete and non-numerical averaging is not applicable, majority voting was used to determine the final prediction. The best performing 5 transfer learning models from the previous section were used to form the ensemble method.

The prediction of the *n*th classifier can be defined as *P_n,k_* ∈ (0, 1) such that *n* = 1, 2, 3…, *N* and *k* = 1, 2, 3…, *K*, where *K* is the number of classifiers and *N* is the number of classes. If the *n*th classifier predicts class *w_k_*, then *P_n,k_* = 1, otherwise *P_n,k_* = 0 [26]. This is described by the below Equation (1).
(1)$$\overset{N}{\underset{n=1}{\sum }}P_{n,k}=max_{k=1}^{K}\overset{N}{\underset{n=1}{\sum }}P_{n,k}\;$$

In practice, this ensemble model works by each of the 5 transfer learning models making their own predictions. Following these individual predictions, the final classification is predicted based on the total number of votes for a given class.

### 2.4. Creation of Novel Convolutional Neural Network Models

Convolutional neural networks (CNN) are commonly used in image classification models due to their high accuracy in supervised learning [27]. CNNs are preferred to fully connected neural networks (FCNN) for image classification because they do not utilize connection weights for pixels and instead use kernels, reducing computational complexity and number of required samples [28,29]. 

#### 2.4.1. Convolutional Layer

Convolutional layers are a crucial part of a CNN model. The input of a convolutional layer consists of 1 or more 2D matrices, while the output of the layer is multiple 2D matrices. The equation below describes the method of computing a single output matrix (2).
(2)$$A_{j}=f\left(\overset{N}{\underset{i=1}{\sum }}I_{i}*\;K_{i,j}+B_{j}\right)$$

First, each input matrix *I_i_* is convolved with the corresponding kernel matrix *K_i,j_*. Next, the sum of all of the convolved matrices is calculated and a bias *B_j_* is added to the sum. Finally, an activation function is applied to the resulting matrix to produce the output matrix [29]. 

#### 2.4.2. ReLU Activation Function

There are many different activation functions that can be used, however, many common functions such as the sigmoid and hyperbolic tangent are saturating nonlinear functions. In saturating nonlinear functions, when the input increases, the gradient output rapidly decreases to 0. This creates what is referred to as the vanishing gradient problem [19]. The rectified linear function (ReLU) is a non-saturating non-linear function that avoids this issue. The ReLU function is defined as f(x)=max(0,x), where f(x)=x if x>0, and f(x)=0 otherwise [29].

#### 2.4.3. Max Pooling Layer

Pooling is used to reduce the feature dimensionality by combining neighboring elements in the output matrices of the convolutional layers. All of the max-pooling layers used in this work consist of a 2 × 2 kernel size to select the maximum value of 4 neighboring elements in a matrix and return a singular element to the output matrix [29]. 

#### 2.4.4. Architecture of the CNN Model

The complexity of the images determines the corresponding complexity of the model. Given the high variability within each class and high similarity between certain classes such as HSIL and SCC, an appropriately large number of convolutional layers and fully connected dense layers are needed to be able to accurately distinguish between the 4 classes. 

The model consists of a series of alternating 2D convolutional layers and max-pooling layers. The final two layers consist of fully connected dense layers. The first dense layer consists of 64 dimensions and the second layer consists of 4 dimensions representing the 4 classes. The highest value of the final dense layer, which consists of a 1D array of length 4, indicates the prediction of that model for a given input.

The model was trained for 230 epochs with a learning rate of 0.0005. Slow learning was preferred in this model due to the complexity of the multi-cell images. Slow learning also allowed for size constriction within the model, as previous attempts in the training process with a faster learning rate required a significantly larger model to achieve comparable results. 

Using a smaller model was thus found to be more resource, memory, and time-intensive. The resulting model consisted of 60,772 trainable parameters, which is 99.78% smaller than the average number of parameters of the models with transfer learning. A visual schematic of the model’s complete architecture is provided in Figure 2.

#### 2.4.5. Architecture of the Autoencoder

Autoencoders have many uses for image classification. The method used herein aims to lower the complexity of the matrix representation of the images to reduce the number of trainable parameters used in CNN training. Autoencoders consist of 2 parts, the encoder, and the decoder. These parts are typically designed to be reflections of each other, where the encoder decreases and the decoder increases in dimensionality [30,31]. The aim of the encoder is to create a condensed vectorial representation of each image. Whereas the decoder uses this vector to recreate the most accurate estimate of the original image from significantly less information. 

For this model, both the encoder and decoder were used to train the autoencoder. Next, the training set was passed through just the encoder to obtain the compressed vector representations, which were then used to train the CNN. The autoencoder was trained for 100 epochs at a learning rate of 0.00005. The CNN was trained for 350 epochs at a learning rate of 0.00005.

Given the reduction in model size, it was necessary to train for longer to allow adequate iterations for the model to fully learn to classify the dataset. A dropout layer that omitted 20% of each batch of data was included to reduce the risk of overfitting due to the lower complexity of the data. 

The basic CNN model discussed in the previous section had 60,772 trainable parameters, while the AE CNN model had 20,355 for the AE and 10,292 for the optimized CNN. This is a 50% reduction in the number of trainable parameters between the two approaches. Parameter reduction is necessary in order to limit memory and resource usage. The complete architecture of the autoencoder and CNN are shown in Figure 3B,C, respectively.

### 2.5. Application of Unsupervised Domain Adaptation

Domain adaptation is a technique that is frequently used to improve model performance on target datasets that are not independent and identically distributed when compared to the training dataset [16,32]. Often, classification systems based on deep neural networks struggle to generalize across data with different input distributions [33]. Domain adaptation is a method to compensate for domain shift-based performance degradation. In unsupervised domain adaptation, there is no labeled training data in the target domain, which is a better replication of the application of this type of classifier in the real world.

#### Deep CORAL

Unsupervised domain adaptation was applied in order to examine its utility in the classification of liquid Pap smear images that were significantly different in appearance from the images used to train the model. CORrelation Alignment (CORAL) is a method that minimizes domain shift via the alignment of second-order statistics of source and target distributions [16]. CORAL works by combining a differentiable loss function referred to as the CORAL loss with the standard classification loss during training. CORAL functions without the need for any target labels. Deep CORAL was only used in the testing of the novel, ThinPrep Pap dataset (Figure 4).

The mathematical basis behind Deep CORAL has been described elsewhere in detail [16]. Notably, CORAL loss is defined as distance between the covariances, or second-order statistics, of the features in the training and test data.
(3)$$l_{CORAL}=\frac{1}{4d^{2}}\;∥C_{S}-C_{T}{∥}_{F}^{2}$$

In which *d* represents the dimensions of deep layers. *C_S_* and *C_T_* are representative of the covariance matrices of features for the source and test datasets, respectively. $∥\cdot {∥}_{F}^{2}$ represents the squared matrix Frobenius norm. *C_S_* and *C_T_* can also be written as follows.
(4)$$C_{S}=\frac{1}{n_{S}-1}\left(D_{S}^{⏉}D_{S}-\frac{1}{n_{S}}{\left(1^{⏉}\;D_{S}\right)}^{⏉}\left(1^{⏉}\;D_{S}\right)\right)$$
(5)$$C_{T}=\frac{1}{n_{T}-1}\left(D_{T}^{⏉}D_{T}-\frac{1}{n_{T}}{\left(1^{⏉}\;D_{T}\right)}^{⏉}\left(1^{⏉}\;D_{T}\right)\right)$$

In which 1 is representative of a column vector with elements equivalent to 1. 

Batch covariances are utilized, and the network parameters between the two networks are shared. Classification and CORAL losses are used to simultaneously train the model as minimization of classification loss alone will lead to overfitting and minimization of CORAL loss alone will lead to feature degeneration. The overall loss function used can therefore be represented by:(6)$$l=l_{CLASS}+\lambda \cdot l_{CORAL}$$
where *λ* represents a hyperparameter that is tuned separately for each model.

## 3. Results

In this study, the performance of 5 publicly available deep learning models was evaluated in the classification of clinical liquid Pap smear images according to the Bethesda classification system. Transfer learning was used, as well as an ensemble classifier composed of said models. The performance of this ensemble approach was assessed in comparison to novel CNN models.

A streamlined and optimized CNN was used with and without an autoencoder. Each model was first trained, validated, and tested on the Hussain et al. SurePath liquid cytology dataset. The test set consists entirely of unseen images to prevent the inflation of the accuracy based on previously learned images. 

Performance was assessed by examining a variety of metrics. Training and validation accuracy and loss were assessed for all tested models (Appendix A), and confusion matrices and classification reports were created (Appendix A) in addition to Receiver Operating Characteristic (ROC) curves. Chi-square, *p*-value, and standard deviations were calculated for each model, and model consistency across runs was assessed (Appendix A). Training and testing runtimes were analyzed for each model (Appendix A). Epoch number, learning rate, activation function, dropout, and batch size information were also characterized for each tested model (Appendix A). Finally, each model was evaluated on its ability to generalize outside of the images it was trained on by assessing classification performance on images produced using a different preparation method: ThinPrep. A novel clinical image test set was created using images collected at Women and Infants Hospital prepared with a different set of sample collection and standardized clinical laboratory protocols for Hologic’s ThinPrep Pap test. Domain adaptation was also assessed as a technique to improve results.

### 3.1. Evaluation of Deep Learning Models with Transfer Learning

To determine the best performing deep learning models with transfer learning, all of the publicly available pre-trained deep learning classifiers were first trained and tested. From there, the 5 models with the highest accuracy were selected for further investigation. These models were ResNet50, DenseNet121, ResNet101, ResNet153, and EfficientNetB0 (Figure 5).

ResNet50 was found to have the highest performance of all the assessed transfer learning models at 99.2% and only misclassified 4 out of the 492 images in the test set (Figure 6). Each of the tested transfer models had between 4–59 million trainable parameters, which enabled them to learn complex patterns and features that smaller models might not be able to effectively learn.

#### 3.1.1. Evaluation of Ensemble Method

The ensemble method was created by using the top 5 performing transfer learning models and making a final prediction based on voting. As could be expected, the ensemble method performed similarly to ResNet50. Given that the best performing individual trained method had a 99.2% accuracy and the rest of the models within the ensemble had lower accuracies, it was unlikely for the ensemble method to surpass the accuracy of ResNet50, however, an accuracy of 98.98% was still obtained (Figure 7).

#### 3.1.2. Evaluation of CNN

The performance of a novel streamlined CNN was evaluated and compared to the larger transfer learning models. We aimed to build a model that would be able to perform consistently and at a comparable level to the pre-trained models obtained with transfer learning without the need for millions of trainable parameters. The model achieved 93.90% accuracy after fine-tuning hyperparameters and determining the ideal combination of convolutional layers (Figure 8). From the confusion matrix, it is evident that the model had the most difficulty differentiating between HSIL and SCC given that they are the two most diagnostically and visually similar classes. Due to this difficulty in discerning these 2 classes, 21 of the images in the SCC class were incorrectly classified as HSIL.

#### 3.1.3. Evaluation of AE-CNN

The addition of an autoencoder to reduce the dimensionality of the data prior to training it using a CNN improved the accuracy from 93.90% to 96.54% (Figure 9). The hyperparameters of both the autoencoder and the CNN were fine-tuned to maximize the accuracy. Although the model still has difficulty consistently differentiating between the HSIL and SCC classes, this is a significant improvement from the CNN model. Additionally, this model was trained with 50% fewer trainable parameters than the CNN, making it the smallest model trained in this study.

### 3.2. Comparisson of All Deep Learning Models

Comprehensive performances of all tested deep learning models were assessed using a summary table to compare metrics (Table 3). ResNet50 had the highest performance of all the fine-tuned models, with the ensemble method a close second.

The number of trainable parameters a model has does not appear to have any correlation to the accurate classification of the images. DenseNet121 had the second-highest accuracy of the pre-trained models, however, it has the second-lowest number of trainable parameters. 

The precision, recall, F1, FPR, FNR metrics are included in the analysis of deep learning models to ensure that the model is not prioritizing the prediction of the largest class to maintain high accuracy. In those cases, while the accuracy might be high the other metrics will be low. However, since this dataset has been balanced as a part of the augmentation process, the metrics listed below are cohesive.

Despite their small size, the CNN and the AE CNN models performed with an accuracy of over 93%, and the AE CNN model outperformed 3 of the pre-trained models and was comparable to ResNet50. This indicates that models with millions of trainable parameters might not be necessary for this classification task. 

In addition to comparing these different metrics, AUC (Area Under the Curve) ROC (Receiver Operating Characteristics) curves were plotted and analyzed for each deep learning model. This was utilized in order to assess how well the models perform for each respective class. Clinically, this method is used to evaluate a diagnostic tool’s ability to differentiate between healthy (negative class) and unhealthy (positive class) samples by measuring the trade-off between the true positive and false-positive rates.

In this case, 5 of the 8 deep learning models included in this study have an AUC value over 0.97 and FPR under 2%, indicating that they are able to reliably distinguish between negative and positive samples (Figure 10). These models include the CNN, AE CNN, ResNet50, DenseNet121, and the Ensemble. According to Figure 10, ResNet152 and EfficientNetB0 are outliers in the SCC and HSIL classes, signifying that those two models are unable to distinguish those classes, often classifying SCC as HSIL and vice versa.

### 3.3. Evaluation of Model Consistency

There is an inherent randomness to deep learning models that comes from the initial random generation of weights and biases. However, despite the randomness of the weights, we expect models to perform comparably between consecutive runs on the same dataset. Table 4 shows the average accuracy of each model as well as the maximum, minimum, and standard deviation for 5 consecutive runs (Table 4).

While the pre-trained models may have performed well on individual runs, there is a larger difference between the runs with the highest and lowest accuracies. Standard deviations for these models ranged from 11.53–19.21%. Despite the high variation in the individual models, the ensemble model performed more consistently, with a lower standard deviation of 8.14%.

However, when compared to the novel CNN models, the ensemble method has a higher standard deviation. The CNN model had a standard deviation of only 2.293%, while the AE CNN was even lower at 1.178%. The CNN model had a much narrower range and an average accuracy that was similar to its previous performance. The AE CNN model was the most consistent in its performance, with the lowest standard deviation and range, and average accuracy that was only 1.45% lower than its performance in the previous section.

### 3.4. Evaluation of Saliency

Two different methods were used to validate the clinical specificity and generalizability of the deep learning models. To address the former, saliency maps were generated, and, to address the later, a novel clinical dataset was created to examine reproducibility of the classification performance for images of a different preparation type. 

Saliency maps are used to highlight the areas that have the most impact on a machine or deep learning model’s final output. This can help to ensure that tested models are learning from the relevant cells and not blank spaces in between them [34]. This is particularly important with deep learning workflows, which tend to be opaquer in terms of feature extraction.

Multiple images from each Bethesda class were examined using two different methods of saliency analysis (Figure 11). The first method is the built-in Saliency method, which uses the gradients of the model’s output instead of the convolutional layer. The second approach utilized is the built-in Keras method [35] ScoreCam++. This method uses the final convolutional layer prior to the dense layers to determine the regions with the most contribution and overlays the original image for comparison. 

Both methods successfully highlight the key features of the images, including cell clusters and individual cells. In particular, visible focal points include areas with abnormal nuclei or different morphologies that characterize the LSIL, HSIL, and SCC states. This is indicative that the model is focusing on the correct areas of each of the 4 classes and that it can therefore be assumed to be more generalizable as it can successfully recognize differences in cells.

### 3.5. Evaluation of Generalizability across Preparation Methods

The first training dataset consisting of images from Hussain et al., was prepared using the SurePath Prep method. While some clinical settings in the United States use SurePath, most have transitioned to using Hologic’s ThinPrep Pap products in more recent years due to various advantages [36,37]. SurePath is still widely used, however, and is significantly more effective than conventional Pap smears [38]. Differences in liquid Pap staining and preparation protocols can produce images that look widely different [39].

In order to evaluate the ability of the SurePath trained model to classify images created with other liquid Pap preparations, and therefore, the potential global utility of the model, images from diagnostically confirmed ThinPrep Pap slides were tested (N = 230). Bethesda and binary classification were both investigated, using a binary classification scheme consisting of NILM + LSIL and HSIL + SCC.

Initial findings indicated that the streamlined CNN AE approach is the most reliable and effective in classifying images from the same preparation method (SurePath), with 96.54% accuracy, only 30,647 trainable parameters, and AUC of 0.975. However, in ThinPrep Pap Bethesda classification, the models consistently overpredict NILM and HSIL and underpredict LSIL and SCC. Images are often misclassified as the adjacent class, such as the misclassification of LSIL as NILM by ResNet101 in Figure 12. Given that all 5 of the pretrained models have the same bias, the ensemble method is unable to compensate for individual biases based on majority voting.

When creating a binary classification scheme to assess model performance at classifying both Bethesda and binary NILM and LSIL or HSIL and SCC, the best results were obtained for ResNet101 (Figure 12). ResNet101 had an AUC value of 0.959 in binary classification and 0.785 in multivariate classification (Figure 13) as well as an accuracy of 88.24% and 61.76% in initial trials without Deep CORAL. Interestingly, the CNN and AE CNN models began performing much less efficiently at classifying these new datasets with the CNN and AE CNN models obtaining AUCs of 0.582 and 0.604, respectively. These were by far the least effective models in cross-classification. It is also notable that, while they performed more or less on par with the other models for NILM, LSIL, and HSIL, the AE CNN and CNN struggled to accurately classify SCC, instead classifying images as HSIL. The ensemble method, however, continued to perform well, with an AUC of 0.975 in initial trials.

It is evident based on Figure 12 and Table 5 that, while the models perform well at classifying images from the same SurePath preparation method, they were not able to translate as well to ThinPrep Pap images. This is evidenced by lower accuracy and precision values, as well as higher false positive and false negative rates.

### 3.6. Evaluation of Deep CORAL’s Efficacy in Cross-Preparation Classification

The next question to investigate was whether the incorporation of Deep CORAL into the processing workflow would improve model classification accuracy when processing this new, dissimilar set of data. All models experienced a consistent increase in performance across multiple metrics, including accuracy (Figure 14; Table 6). 

ResNet101 continued to be the most accurate model in cross-preparation classification in these tests as well. In particular, it had an accuracy of 92.65%, precision of 93.27%, recall of 92.02% and F1 of 92.45%.

When examining the false positive and false negative rates, there was a slightly higher false positive prediction, which can be expected, especially in binary classification schemes in which LSIL is grouped with NILM. In particular, when applying Deep CORAL, ResNet101 had a false positive rate of 13.33% and a false negative rate of 2.63%. The AUC for ResNet101 was 0.986.

While the CNN and AE CNN models improved with Deep CORAL, their performances still remained lower than all other models, consistent with the initial findings on testing the new dataset. All other models had accuracies all above 80%. High classification efficacy that was consistent in a narrow range was observed across all other models in binary classification schemes. More variability was observed with multivariate classification, as is shown in Figure 14.

While some of the tested models can be considered clinically viable in multi-cell SurePath Prep image classification tasks, future work must be conducted in order to optimize model performance across different preparations. This is a useful avenue to pursue in order to create a robust model that can correctly classify numerous liquid Pap preparations.

## 4. Discussion

The use of clinical decision support systems is becoming more common to aid pathologists in diagnosing diseases based on images [40]. Due to the high level of accuracy and precision needed for manual or image-assisted slide screening, maximum workload limits are placed on cytotechnologists to reduce human error. This limit varies between 100–200 slides in no less than an 8-h workday, depending on manual review versus image device-assisted review [41]. Therefore, further development of automated classification tools may significantly improve and streamline clinical workflows.

These findings demonstrate that a deep convolutional neural network is able to accurately classify 4 Bethesda classes of cells in multicell images of SurePath Pap slides using the Hussain et al. dataset. An ensemble of the top 5 pre-trained models was able to achieve an accuracy of 99.18%. Additionally, the much shallower AE CNN model was able to achieve an accuracy of 96.54% while using 99.89% fewer trainable parameters.

Further testing demonstrated that, despite these high-performance metrics, the pre-trained models with transfer learning do not reliably predict as accurately as the novel AE CNN and CNN models. Significant variability is observed in the accuracy of the pre-trained models between 5 consecutive training sessions while the AE CNN model performed more consistently and had a smaller standard deviation. Despite the high variability of the individual pre-trained models, the majority voting used by the ensemble method compensated for low-performing models. Traditionally, larger pre-trained models with transfer learning are used to train smaller datasets to avoid the risk of overfitting, however, some conflicting research suggests that smaller models can also reduce overfitting and improve generalizability [24].

Next, the SurePath trained model was tested with respect to its applicability in classification of the novel ThinPrep dataset. This helped to shed light on the utility of the model across various types of liquid Pap preparations. This will determine the utility of this approach in different settings across the world, where many different liquid Pap preparations are used. While SurePath is a common preparation method, particularly outside of the United States, most clinicians in the United States use Hologic’s ThinPrep products [42]. The ThinPrep 2000 and 5000 loading systems gently disperse samples using a rotary drive mechanism, monitor cell collection using a pneumatic and fluidic system, and transfer cells to the slide with positive air pressure. While this results in more uniformly prepared slides than other methods such as Cytospin, these instruments can retail for tens of thousands of dollars [43].

Portable and other cytological centrifuges are a fraction of this price, but often result in non-uniform dispersal, often depositing rings of cells around the edges of the slide [43,44]. BD’s SurePath system can be used with automated accessories, such as the BD PrepMate, that are similar to the ThinPrep system. However, Hussain et al. and others use cassette-based centrifugation, which can help to reduce costs.

It is also notable that many different Pap staining protocols exist. Some approaches involve automated cell staining, while others involve manual staining. Different staining protocols can result in different appearances of cells [45].

The differences in the appearances of the images between the two preparation methods introduced a high degree of variability that resulted in reduced classification efficacy across all tested deep learning algorithms. The incorporation of unsupervised domain adaptation was able to significantly improve accuracy and other performance metrics. ResNet101 performed the best at classifying the novel ThinPrep dataset, with an accuracy of 92.65%, while the novel CNN and AE CNN accuracies were notably lower.

Differences in staining, magnification and cell density have a significant impact on the appearance of images. In this case, the SurePath dataset was stained and prepared differently from the ThinPrep dataset. Additionally, different staining appearances were observed within the SurePath dataset. Some images within the dataset also appeared to be taken at different magnifications. Of note, the Hussain, et al. dataset images and classifications were used as originally classified, despite discrepancies noted with the dataset’s classification accuracy. Variations within the training set and fundamental differences between SurePath and ThinPrep slides resulted in reduced efficacy when testing the novel dataset.

## 5. Conclusions

In conclusion, deep learning is a promising tool for the classification of liquid Pap smear images. When investigating the ability of SurePath trained models to classify SurePath images, high classification accuracy was obtained for individual transfer learning models, such as ResNet50, as well as ensemble methods, CNN, and AE CNN models. The novel CNN and AE CNN had greater reproducibility than the transfer learning and ensemble approaches. When investigating the potential of SurePath trained models to classify novel ThinPrep images, performance decreased, but results improved with the use of Deep CORAL unsupervised domain adaptation.

Further work must be conducted in this area in order to create larger and more clinically accurate, publicly accessible datasets for preparation methods such as ThinPrep Pap. From the deep learning perspective, we must also analyze whether cross-preparation method classification can be performed by optimizing training and design of improved classification algorithms, and through further pursuit of domain adaptation techniques.

## Figures and Tables

**Figure 1 diagnostics-12-01838-f001:**
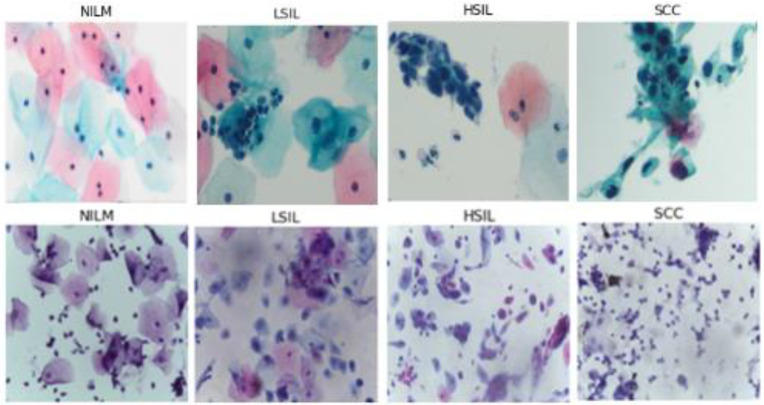
Comparison of sections of images from our novel ThinPrep Pap dataset (above) and the Hussain et al., SurePath Pap dataset (below).

**Figure 2 diagnostics-12-01838-f002:**
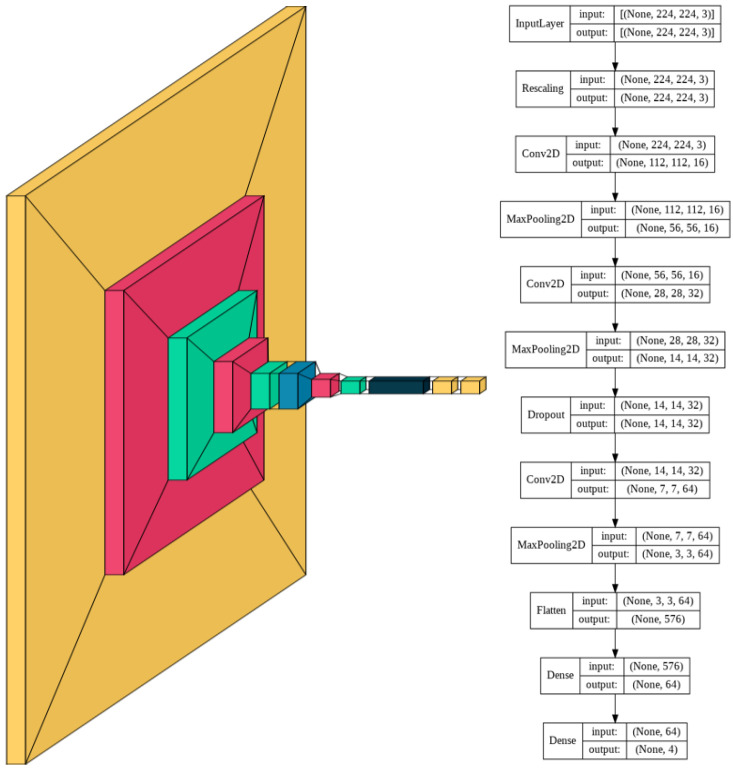
CNN model architecture.

**Figure 3 diagnostics-12-01838-f003:**
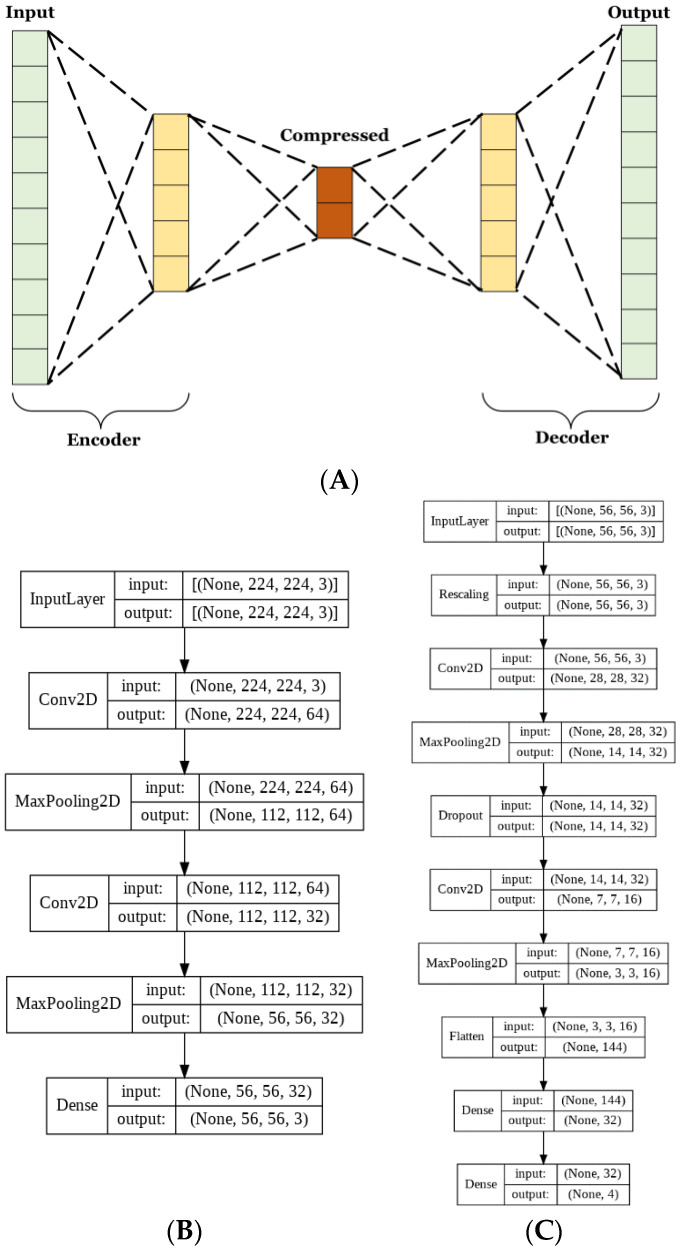
Illustration of (**A**) image compressing autoencoder, (**B**) autoencoder architecture, and (**C**) CNN architecture used in the AE CNN.

**Figure 4 diagnostics-12-01838-f004:**
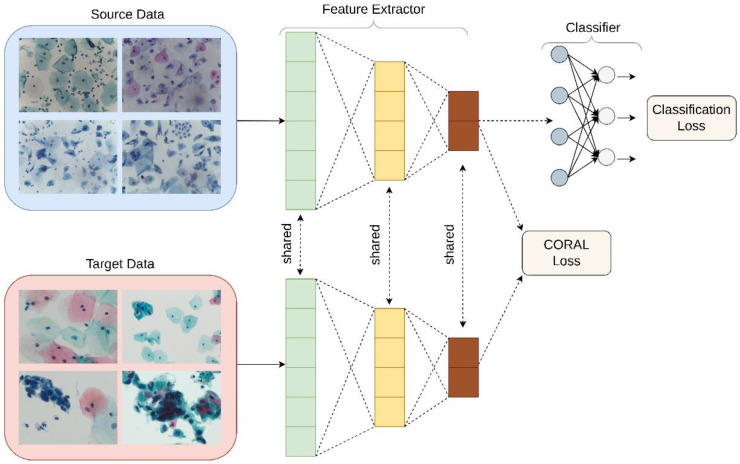
Schematic of Deep CORAL’s integration into the automated classification workflow. Note that “Source Data” represents the SurePath training dataset, while “Target Data” represents the ThinPrep testing set, in this context.

**Figure 5 diagnostics-12-01838-f005:**
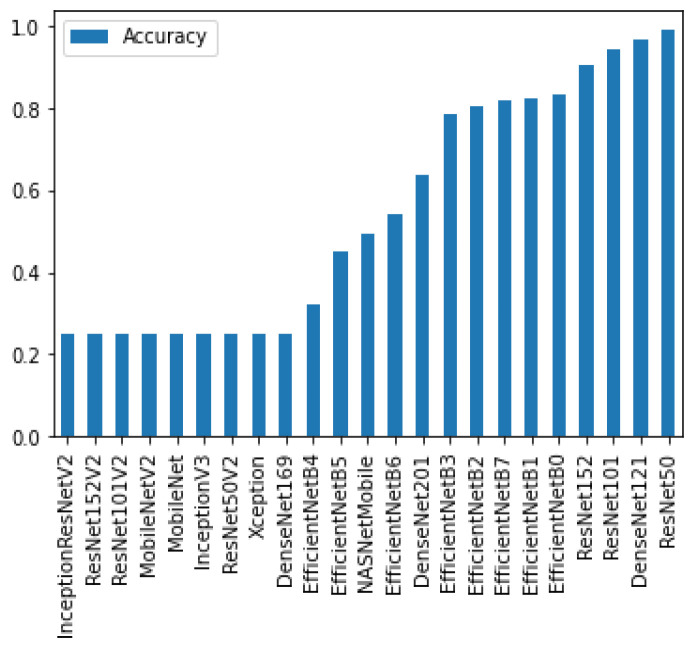
The accuracies of all pre-trained models after training for 25 epochs.

**Figure 6 diagnostics-12-01838-f006:**
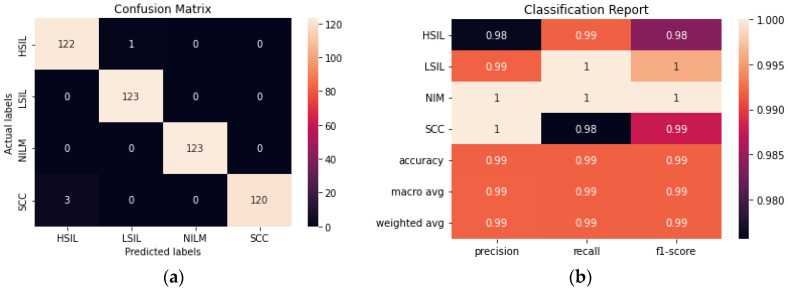
(**a**) Confusion matrix for ResNet50. (**b**) Classification report for ResNet50.

**Figure 7 diagnostics-12-01838-f007:**
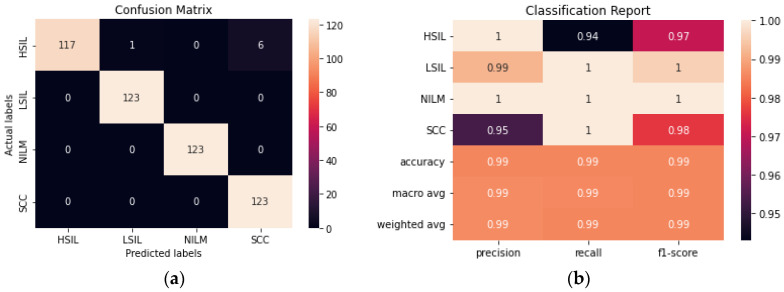
(**a**) Confusion matrix for Ensemble. (**b**) Classification report for Ensemble.

**Figure 8 diagnostics-12-01838-f008:**
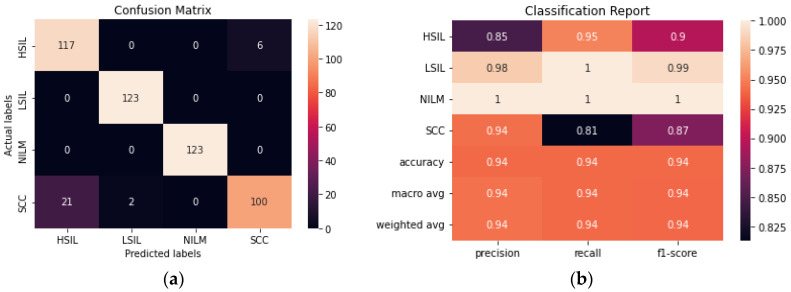
(**a**) Confusion matrix for CNN. (**b**) Classification report for CNN.

**Figure 9 diagnostics-12-01838-f009:**
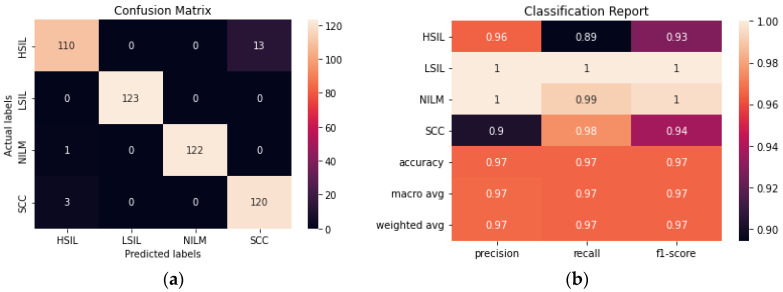
(**a**) Confusion matrix for AE-CNN. (**b**) Classification report for AE-CNN.

**Figure 10 diagnostics-12-01838-f010:**
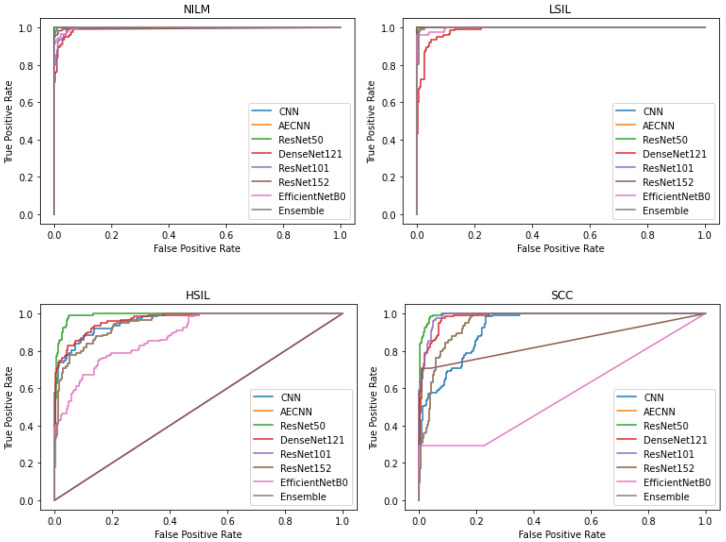
Comparison of AUC-ROC curve across all models.

**Figure 11 diagnostics-12-01838-f011:**
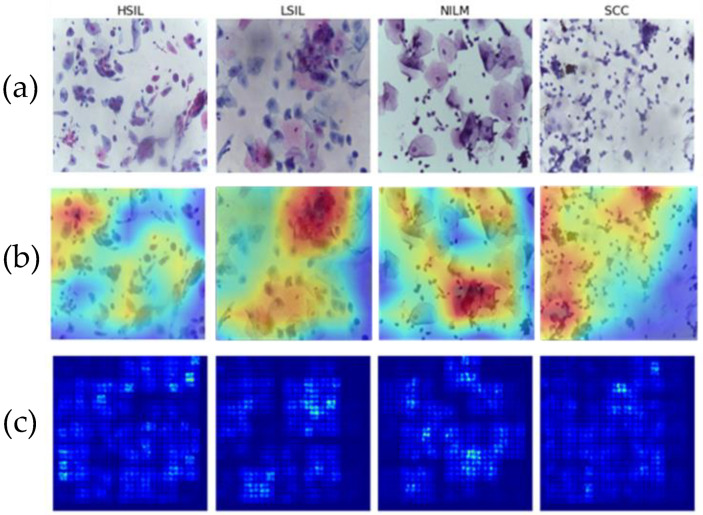
Visualizing model learning on (**a**) representative images for each Bethesda class (**b**) using saliency maps and (**c**) ScoreCam++.

**Figure 12 diagnostics-12-01838-f012:**
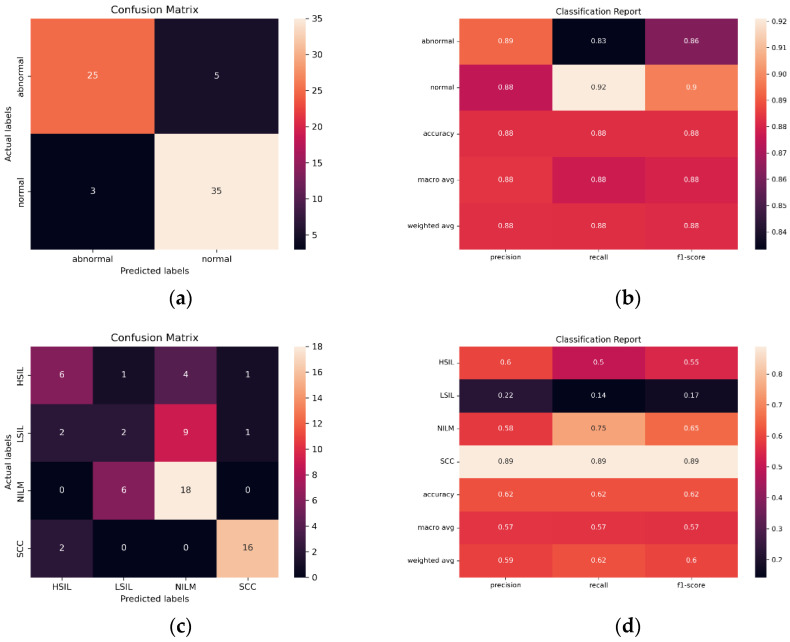
(**a**) Binary confusion matrix and (**b**) classification report for the best performing classifier, ResNet101 when using the novel ThinPrep Pap image test set. (**c**) Multivariate confusion matrix and (**d**) classification report for ResNet101 when using the novel ThinPrep Pap image test set.

**Figure 13 diagnostics-12-01838-f013:**
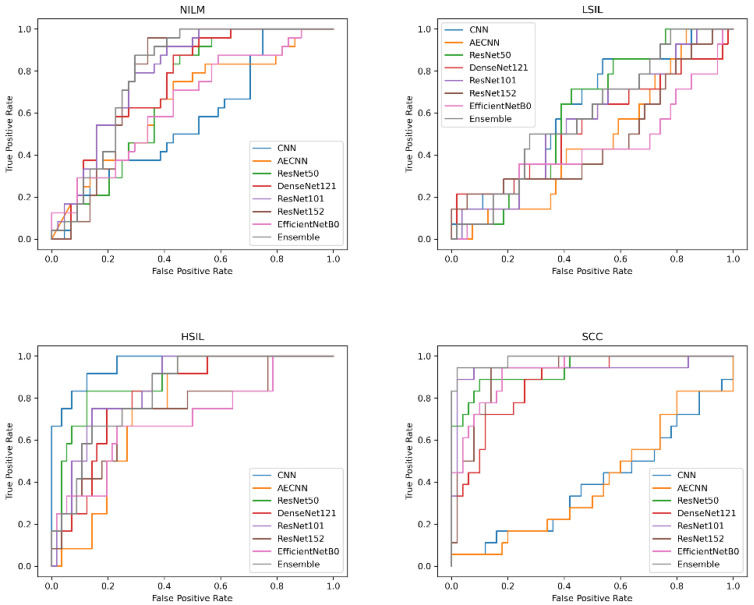
AUC-ROC curves obtained using the novel ThinPrep Pap image test set.

**Figure 14 diagnostics-12-01838-f014:**
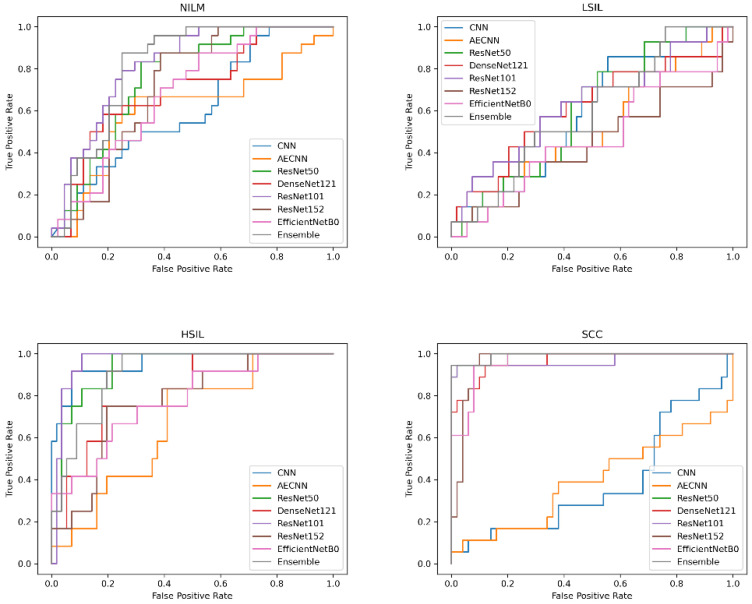
AUC-ROC curves obtained using the novel ThinPrep Pap image test set.

**Table 1 diagnostics-12-01838-t001:** Number of samples in each class for the original and augmented SurePath datasets, as well as the number of images in the novel, ThinPrep Pap dataset.

Class	Original Number of SurePath Images	Augmented Number of SurePath Images	Number of ThinPrep Pap Images
NILM	613	613	72
LSIL	113	613	56
HSIL	163	613	48
SCC	74	613	54
**Total Images**	**963**	**2452**	**230**

**Table 2 diagnostics-12-01838-t002:** Number of trainable parameters for top 5 transfer learning models.

Model Name	No. Trainable Parameters
ResNet50	23,587,712
DenseNet121	7,037,504
ResNet101	42,658,176
ResNet152	58,370,944
EfficientNetB0	4,049,571

**Table 3 diagnostics-12-01838-t003:** Comparison of fine-tuned models.

Models	No. Trainable Parameters	Accuracy	Precision	Recall	F1	FPR	FNR	AUC
**Basic CNN Model**	60,772	93.90%	94.42%	93.90%	93.86%	2.03%	1.96%	0.972
**AE CNN**	**30,647**	96.54%	96.68%	96.54%	96.54%	1.15%	1.14%	0.975
**ResNet50**	23,587,712	**99.19%**	**99.20%**	**99.19%**	**99.19%**	**0.27%**	**0.27%**	**0.996**
**DenseNet121**	7,037,504	96.55%	96.65%	96.55%	96.57%	1.16%	1.17%	0.982
**ResNet101**	42,658,176	94.51%	95.04%	94.51%	94.34%	1.90%	1.72%	0.871
**ResNet152**	58,370,944	90.65%	91.28%	90.65%	90.48%	3.18%	3.02%	0.835
**EfficientNetB0**	4,049,571	83.13%	86.02%	83.13%	82.51%	6.27%	5.74%	0.861
**Ensemble**	-	98.98%	98.99%	98.98%	98.98%	0.34%	0.34%	0.994

**Table 4 diagnostics-12-01838-t004:** Comparing average accuracy of fine-tuned models for 5 consecutive runs.

Model	Average Accuracy	Standard Deviation	Max	Min
**CNN**	91.28%	2.293%	94.28%	88.21%
**AE CNN**	95.10%	1.178%	96.51%	93.29%
**ResNet50**	74.72%	17.45%	99.59%	47.97%
**DenseNet121**	80.61%	18.96%	95.33%	46.14%
**ResNet101**	64.84%	19.21%	96.75%	41.67%
**ResNet152**	77.28%	11.53%	99.19%	64.84%
**EfficientNetB0**	80.89%	19.05%	97.76%	48.58%
**Ensemble**	88.67%	8.14%	97.97%	73.45%

**Table 5 diagnostics-12-01838-t005:** Comparison of SurePath dataset trained models’ performance on classification using the novel ThinPrep Pap dataset.

Model	Accuracy	Precision	Recall	F1	FPR	FNR	AUC
**CNN**	60.29%	60.29%	57.81%	56.93%	63.34%	21.05%	0.582
**AE CNN**	58.82%	58.82%	59.30%	58.79%	36.67%	44.74%	0.604
**ResNet50**	83.82%	83.82%	82.37%	83.00%	30.00%	5.26%	0.904
**DenseNet121**	75.00%	75.00%	72.02%	71.77%	53.34%	2.63%	0.928
**ResNet101**	88.24%	88.24%	87.72%	87.98%	16.67%	7.89%	0.959
**ResNet152**	85.29%	85.29%	84.74%	84.97%	20.00%	10.53%	0.904
**EfficientNetB0**	77.94%	77.94%	75.35%	75.62%	46.67%	2.63%	0.882
**Ensemble**	82.35%	82.35%	80.00%	80.68%	40.0%	0.00%	0.975

**Table 6 diagnostics-12-01838-t006:** Comparison of SurePath dataset trained models’ performance on classification using the novel ThinPrep Pap dataset when adding in Deep CORAL into the Processing Workflow.

Model	Accuracy	Precision	Recall	F1	FPR	FNR	AUC
**CNN**	66.18%	60.29%	63.42%	62.61%	60.00%	13.16%	0.600
**AE CNN**	60.29%	58.82%	61.32%	60.29%	30.00%	47.37%	0.530
**ResNet50**	89.71%	83.82%	88.33%	89.18%	23.33%	0.00%	0.970
**DenseNet121**	83.82%	75.00%	81.67%	82.45%	36.67%	0.00%	0.951
**ResNet101**	92.65%	88.24%	92.02%	92.45%	13.33%	2.63%	0.986
**ResNet152**	86.76%	85.29%	85.35%	86.09%	26.67%	2.63%	0.940
**EfficientNetB0**	80.88%	77.94%	79.04%	79.61%	36.67%	5.26%	0.918
**Ensemble**	85.29%	82.35%	83.34%	84.19%	33.34%	0.00%	0.984

## Data Availability

All data presented in this study will be made available upon reasonable request through contact of the supporting author.

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
