# Peer review of "A Comparative Analysis of Deep Learning Models for Automated Cross-Preparation Diagnosis of Multi-Cell Liquid Pap Smear Images"

_diagnostics, 2022, doi:10.3390/diagnostics12081838_

Round 1

Reviewer 1 Report

The authors have presented a comparative analysis of deep learning models for cross-preparation diagnosis of Multi-Cell Liquid Pap Smear Images. There are several points that the authors should address in this manuscript which are listed below:

·        On page 3 it’s written “Additional images for training were created using the ImageDataGenerator function 105 from the Keras library” This sentences lack a reference(s)

·        Network hyperparameters are preferably to be inserted in a sperate table

·        Section (2.4. Creation of Novel Convolutional Neural Network Models) has a text book information. What is the point of inserting this sub-section?

·         More details about the structure of this model (3.1.1. Evaluation of Ensemble Method) should be inserted. Although the authors has given some details in section 2.3 but the given information is somehow shallow

·      The authors have mainly compared their work with Hussain et al. More related work should be included to make the work comprehensive

·        As the main purpose of the work is to compare different deep learning model, including the timing performance and/or complexity is a must

Author Response

The authors have presented a comparative analysis of deep learning models for cross-preparation diagnosis of Multi-Cell Liquid Pap Smear Images. There are several points that the authors should address in this manuscript which are listed below:

On page 3 it’s written “Additional images for training were created using the ImageDataGenerator function 105 from the Keras library” This sentences lack a reference(s)

-We thank you for bringing this to our attention. We have included a citation, #37, that redirects to this source.

Network hyperparameters are preferably to be inserted in a sperate table

-Thank you for pointing this out. Network structure has been described in a table which we have added to the Supporting Documents, Supplemental Figure 10. 

-This figure in particular has information on epoch, learning rate, activation function, dropout, and batch size.

-We have also aimed to visually illustrate the layering and composition of the models we have created in Figure 1 and Figure 2, and have described certain parameters for testing in the Methods section.

-Please let us know if there is anything else that should be added or if this will be sufficient.

Section (2.4. Creation of Novel Convolutional Neural Network Models) has a text book information. What is the point of inserting this sub-section?

-We thank you for your comment. We have heavily edited this section to be more concise and less cursory. The section has been simplified from several sentences down to 2 sentences.

-If you feel that we should remove the section completely following this edit, please let us know and we will do so!

More details about the structure of this model (3.1.1. Evaluation of Ensemble Method) should be inserted. Although the authors has given some details in section 2.3 but the given information is somehow shallow

-We thank you for your comment. We have included more information on the ensemble method in the methods section. In addition to the provided equation, we have added a verbal description of how the ensemble classification works, which we believe will be helpful.

-We have also added some context to this section in the results that you have mentioned explaining again what the method is and how it works.

-We believe that this will increase the clarity of how the model works. If that is not the case, please let us know.

The authors have mainly compared their work with Hussain et al. More related work should be included to make the work comprehensive

-We thank you for pointing this out, we have added in more relevant references to the paper, which we have incorporated for the purpose of strengthening our introduction and building our paper on the foundation of existing Deep Learning Pap Smear work.

As the main purpose of the work is to compare different deep learning model, including the timing performance and/or complexity is a must

-We thank you for this comment. We have attached information of this nature in the Supporting Documents section, in Supplemental Figure 9.

-This figure highlights training runtime across all the samples, as well as testing runtime per individual sample. We hope that this will provide some context in this area.

-We thank you for all of your comments. We have made several edits to the text to the existing material that was included for your initial review which we believe will make the paper more suitable for publication and address your concerns. 

-We have also performed significant additional work for this revision, which we believe will be well received and will strengthen the paper. In particular, we added more images to the novel dataset we created for testing. We refined our model and even included an entire new section of the paper focusing on domain adaptation, in which we applied a method called Deep CORAL to examine how to improve cross-preparation classification accuracy. We achieved significantly better results during this process, and have now been able to achieve an accuracy of 92.65%, a major improvement from our former 75% highest accuracy for the second section of the paper. We have added significant information to the methods, results, and discussion, including many new equations and figures that we feel will be helpful.

Reviewer 2 Report

This paper is trying to discuss the efficiency of each deep learning model for the diagnosis of multi-cell liquid pap smear images and it may help to guide people toward a better analysis model. However, the paper does not show significant advantage of one model from the others. Does the discussion for these models really necessary for future work? In addition, the data sets are small. The calculation based on a small set of data is not significant enough. 

Author Response

-We thank you for all of your comments. We have made several edits to the text to the existing material that was included for your initial review which we believe will make the paper more suitable for publication and address your concerns. We will also address your written comments point by point:

-You are correct that, with different training and parameter use, all of the models were able to perform quite well in the first round of experiments. However, we frame the novel CNN and AE CNN models as a more streamlined approach to classification as they have few parameters. While the transfer and ensemble models have many more parameters, they take less training epochs to be trained in classifying the images. We present here a way to achieve successful outcomes with transfer learning, ensemble methods, or smaller CNN models. While transfer learning approaches may be desirable to some researchers, they will likely be too bulky or time consuming for other researchers, who may be interested in a bespoke CNN approach. Therefore, we feel that the discussion and comparison of all of these models is necessary to inform future work by a variety of scientists, and that performances should be considered alongside the different parameters and requirements of each model.

-We believe that this is particularly evident with our new results - the novel CNN and AE CNN are streamlined and perform very well in the first section of the paper, with classifying images of the same preparation type. However, in the second section of the paper, the transfer learning (more complex) models are better able to bridge the gap between the two different sets of data.

-We have been inspired by your comments to aim to clarify how the models are different, particularly in the second section of the paper, where you can see more distinct patterns.

-While you are correct that the tested datasets are small, particularly in the landscape of machine learning papers, the included dataset is the largest publicly available multi-cell liquid pap smear dataset. The second, novel dataset that was tested was not meant to be a comprehensive new dataset and was only used to examine transferability to a different preparation method (ThinPrep Pap) for which there are no publicly available multi-cell datasets.

-Due to your important comment, we have taken the liberty of collecting more images for our dataset, which we have added into the paper.

-We would also like to address your feedback that the calculation based on the second, novel dataset is not significant enough. We have mainly included this test because there is very little literature examining the transferability of deep learning beyond the 1 training set, which is likely because other researchers have also obtained results that are not as impressive. We feel this is an important area to explore, however. We have performed significant additional work to add into the paper to explain more of our direction.

-We refined our model and even included an entire new section of the paper focusing on domain adaptation, in which we applied a method called Deep CORAL to examine how to improve cross-preparation classification accuracy. 

-With this approach, we go beyond showing in the second part of the paper that classification is less accurate, we have actually been able to improve it and show some possible ways that others can tackle this challenge. We think this will make the paper more useful to the field.

-We achieved significantly better results during this process, and have now been able to achieve an accuracy of 92.65%, a major improvement from our former 75% highest accuracy for the second section of the paper. We hope that this will be in line, or at least closer to, the expectations of yourself and the journal.

Round 2

Reviewer 1 Report

The authors have addressed my previous concerns. The authors have included some of the responses in the supplementary file. If this supplementary file is only available to the reviwers then this information is better to be included within the manuscript as well as it gives insights to the future readers. (especially the supplementary information of figure 10)

Author Response

-We thank you for these additional comments. We have reviewed the paper once more and made many changes to the writing style and grammar in order to address your feedback that moderate English changes are required. We believe this will improve the language and readability of the paper.

-We also thank you for the additional comment on the supplementary information and call outs to it in the main paper. The supplementary information will be published along with the paper and will be visible to all readers. We have also included a call out and reference to each Supplemental file and figure in the main body of the text. We believe that this will address your concerns.

-In particular, we have added in the following paragraph to the main text in order to address this point “Performance was assessed by examining a variety of metrics… (Supplemental Figure 10)”.

Reviewer 2 Report

The paper is qualified to be published on Diagnostics.

Author Response

We thank you for this comment and for your work with reviewing our manuscript.